

# Drift-aware sea ice thickness maps from satellite remote sensing

Robert Ricker[1], Thomas Lavergne[2], Stefan Hendricks[3], Stephan Paul[3], Emily Down[2], Mari Anne Killie[2], and Marion Bocquet[1]

[1]NORCE Norwegian Research Centre, Tromsø, Norway
[2]Norwegian Meteorological Institute, Oslo, Norway
[3]Alfred Wegener Institute, Helmholtz Centre for Polar and Marine Research, Bremerhaven, Germany

**Correspondence:** Robert Ricker (rori@norceresearch.no)

**Abstract.** The standard approach to derive gridded sea ice thickness (SIT) is to aggregate the original along-track estimates from satellite altimeters over a one-month period. However, this approach neglects processes like sea ice advection, deformation, and thermodynamic growth that occur within the aggregation period. To address these limitations, we propose a drift-aware method that accounts for sea ice motion and SIT changes due to dynamics and thermodynamics in monthly SIT products. We present a method to derive daily drift-aware sea ice thickness (DA-SIT) maps for the Arctic, based on Envisat and CryoSat-2 along-track data. The approach is validated against buoys, airborne SIT surveys and moored upward-looking sonar (ULS) measurements. DA-SIT demonstrates the ability to register sea ice thickness anomalies, which are also observed by daily ULS SIT averages, while being overlooked by the conventional gridded SIT data. Comparative analysis reveals that DA-SIT reduces orbit trackiness patterns and improves consistency in regions with significant ice drift, such as the Transpolar Drift. The drift-awareness enables detailed studies of regional sea ice dynamics and fluxes, while improving co-registration of multi-mission satellite data. However, when considering pan-Arctic estimates of ice volume, we do not expect significant changes in time series and trends compared to existing studies.

## 1 Introduction

The polar regions are a hot spot of climate change, associated with rising air temperatures (Landrum and Holland, 2020) and the transition of the Arctic Ocean to a state more closely to Atlantic waters, a process known as Atlantification (Polyakov et al., 2017). As a consequence, sea ice significantly declined during the last decades, both in extent and thickness (Comiso et al., 2008; Lindsay and Schweiger, 2015; Ricker et al., 2021). To monitor sea ice decline on large scale, satellite observations are crucial. Observing sea ice thickness (SIT) is important as its magnitude regulates the heat exchange between ocean and atmosphere. In fact, SIT is an Essential Climate Variable (ECV) quantity (Lavergne et al., 2022) and critical to calculate sea ice volume and mass balance (Bocquet et al., 2024) as well as fresh water fluxes (Ricker et al., 2018; Selyuzhenok et al., 2020).

Satellite altimetry has become the major tool for estimating SIT (Laxon et al., 2013; Ricker et al., 2014; Tilling et al., 2018; Petty et al., 2020). Within the European Space Agency (ESA) Climate Change Initiative (CCI), consistent SIT time series are developed across multiple satellite radar altimetry missions (ERS, Envisat, CryoSat-2, and Sentinel-3) to observe long-term trends (Paul et al., 2018). However, existing methods for producing monthly SIT maps suffer from the fact that the sea ice



is constantly in motion (Spreen et al., 2011). With the standard approach, along track SIT estimates derived from satellite altimetry measurements are typically collected over one month, allowing for an equal orbit coverage of the polar regions (Sallila et al., 2019). SIT is then averaged onto a fixed grid, where each input SIT estimate is geolocated at the time of the original measurement. Other methods use more advanced interpolation techniques, such as optimal interpolation or Bayesian approaches (Ricker et al., 2017; Gregory et al., 2021). All these approaches neglect sea ice drift occurring within the data

collection period, and therefore introduces uncertainties in the final SIT maps. In particular, we will show that neglecting sea ice drift causes significant spatial blurring and non-uniform temporal coverage in general, compromising the accuracy of regional thickness distributions. This is also important in the context of increasing sea ice drift speed over the last decades, and studies suggest that sea ice mobility will further increase in the future (Kwok et al., 2013; Zhang et al., 2022). In some regions, like in the Transpolar Drift or Fram Strait, the geodesic displacement of sea ice can reach several 100 kilometers within a

month (Kwok et al., 2004; Lavergne and Down, 2023). In addition to the advection of sea ice, processes like deformation and thermodynamic ice growth occur within the data collection period of one month, meaning that SIT of a certain parcel derived in the beginning of a month will be altered by the end of the collection period. Moreover, because the temporal distribution of satellite orbits is uneven across regions throughout the month, the uncertainty associated with SIT changes will vary spatially.

    While sea ice drift is increasing, satellite altimeter technology advances, and sensors such as the laser altimeter on the NASA

Ice, Cloud, and land Elevation Satellite 2 (ICESat-2) are sensitive enough to even register small SIT anomalies (Petty et al., 2023). Without taking into account sea ice drift, this information might get lost when applying standard gridding methods to generate monthly SIT maps. And when obtaining geophysical information from multi-platform data, a correction for ice motion before data merging is a strong candidate for accuracy improvement. For example, the CRYO2ICE campaign (Fredensborg Hansen et al., 2024) to observer snow depth on sea ice by ESA's CryoSat-2 radar altimeter and NASA's ICESat-2

altimeter currently relies on single synchronized orbits due to sea ice drift.

    To reduce uncertainties in SIT maps due to sea ice drift, we propose a novel approach that incorporates sea ice drift estimates from passive microwave satellite radiometers and combines them with satellite altimetry data to derive drift-aware sea ice thickness maps. This method allows individual parcels of satellite altimeter measurements to be advected over time, correcting for motion while also capturing sea ice thickness change as a consequence of thermodynamic growth and deformation effects

between satellite overpasses.

    Therefore our objective is to describe the drift-aware sea ice thickness (DA-SIT) algorithm, based on data from Envisat (2002-2012) and CryoSat-2 (2010-2020). In addition, our goal is to validate and benchmark DA-SIT using drifting buoys, airborne sea-ice thickness surveys and moored upward looking sonars (ULS), while assessing the impact of the drift-awareness algorithm by a comparison with conventionally gridded sea ice thickness products.

This paper is outlined as follows: in Section 2 in the first part, we describe the individual input data sets, while in the second part, we describe the drift-aware processing algorithm, including the associated uncertainty estimation. In Section 3, we present the results from the DA-SIT evaluation against independent validation datasets. In Section 4, we explore the impact of DA-SIT. Finally, conclusions are drawn in Section 5.



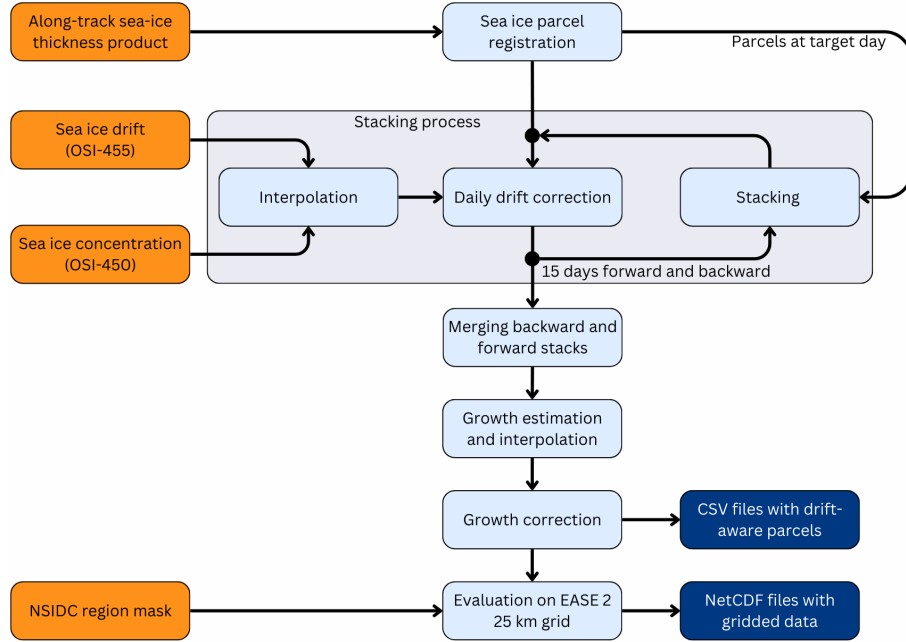

**Figure 1.** Drift-aware sea-ice thickness processing flow chart. Orange boxes indicate input data, light blue boxes indicate processes and dark blue boxes indicate output data products.

## 2 Data and Methods

### 2.1 Input Data

#### 2.1.1 Altimeter Datasets

For the DA-SIT processing, the input altimetry data set contains the along-track geophysical variables including sea-ice thickness (SIT) and corresponding point-wise geolocation information (longitude, latitude). Moreover, additional variables such as snow depth and freeboard are tracked along with the SIT data. Along-track data are typical level-2 datasets.

In this study, we use the latest ESA CCI climate data record version 3.0 of SIT for the northern hemisphere polar region, derived from two satellite missions. For October 2002 to March 2012, we use daily along track SIT data from the Radar Altimeter-2 (RA-2 ) instrument on the Envisat satellite. For November 2010 to April 2020, we use daily SIT data from the SAR Interferometer Radar Altimeter (SIRAL) instrument aboard CryoSat-2. Both datasets cover the winter months (October to April) and are provided at full sensor resolution (Hendricks et al., 2024a, b). This study is a proof-of-concept for single-altimeter data and in principle applicable to other platforms, such as ERS-2, Sentinel-3 and ICESat-2.



### 2.1.2 Sea Ice Concentration

For sea ice concentration, we use the Ocean and Sea Ice Satellite Application Facility (OSI SAF) Global Sea Ice Concentration Climate Data Product (CDR) version 3 (OSI-450-a, 2022), which covers the period 1978 - 2020 and is routinely extended by the Interim CDR OSI-430-a. OSI-450-a is a full reprocessing of sea-ice concentration, using improved algorithms and an
upgraded processing chain (Lavergne et al., 2023, 2019). Sea ice concentration is obtained from passive microwave radiometer data, including the scanning multichannel microwave radiometer (SMMR), the special sensor microwave/imager (SSM/I), and the special sensor microwave imager/sounder (SSMIS), as well as ERA5 reanalysis data. Ice concentration is provided on daily EASE2 grids with a spatial resolution of 25 km.

### 2.1.3 Sea Ice Drift

For sea ice drift, we use the OSI SAF low-resolution Global Sea Ice Drift CDR version 1 (OSI-455, 2022), which covers the period 1991 - 2020. OSI-455 provides displacements over a time span of 24 hours on a 75 km grid and is based on passive microwave radiometer data including satellite sensors SSM/I, SSMIS, the advanced microwave scanning radiometer for Earth Observing System (AMSR-E), and the advanced microwave scanning radiometer 2 (AMSR2). Displacements are estimated using a cross-correlation method on pairs of satellite images. OSI-455 builds on the methodologies of the near-real-time sea
ice drift product OSI-405 (OSI-405, 2007). More details about the used algorithm can be found in Lavergne and Down (2023). If data on individual days are missing, the data product of the closest day available is used. OSI-455 also provides uncertainty estimates (one standard deviation) for the dX and dY components of the drift vector, which are used for the DA-SIT uncertainty estimation.

## 2.2 Drift-Aware Sea Ice Thickness Trajectories

Fig. 1 shows the processing scheme for the drift-aware sea-ice thickness (DA-SIT) product. The processing is divided into several steps, which are executed sequentially for each winter season. In the following we describe the individual steps required to compute DA-SIT, corresponding to Fig. 1.

### 2.2.1 Sea Ice Parcel Registration

In the first step, the along track SIT data products (Envisat, CryoSat-2) are collected on a daily basis. For the sea-ice parcel
registration, these individual data points are aggregated within circular parcels with a radius of $R = \sqrt{2} \cdot 10 \, \text{km}$. This registration in parcels is done to reduce computational costs of the drift correction step. The parcel-wise tracking is also justified considering the coarse spatial resolution of the drift product (see Section 2.1.3).

The spacing between parcel centers is 10 km in both x and y directions, using the NSIDC EASE-Grid 2.0 (Brodzik et al., 2012). The chosen radius ensures partial overlap between parcels, improving the robustness of SIT distribution representation
and mitigating geolocation errors.



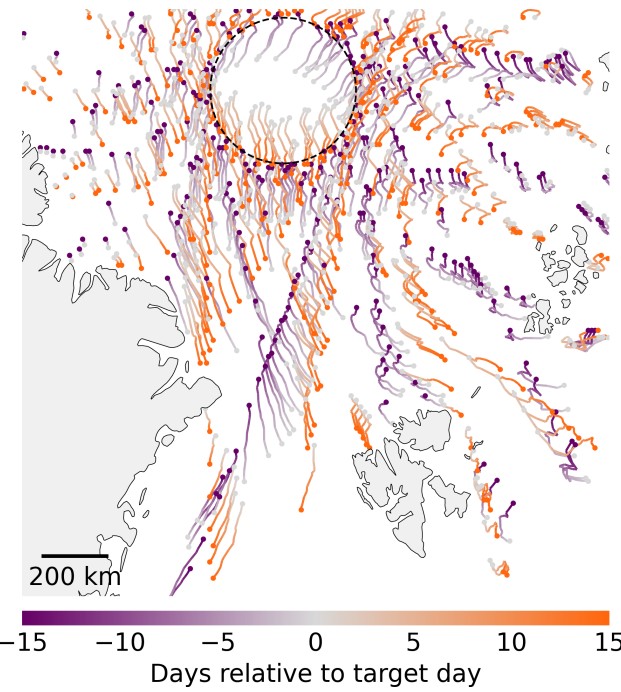

**Figure 2.** Every 10th sea ice parcel trajectory with origin in beginning or end of March 2020, and target day on the 15th of March. The black dashed circle at 88°N marks the CryoSat-2 pole hole.

Parcels are initially registered using the centroid of each grid cell, corresponding to the central position of each circular polygon. The mean SIT is calculated for each parcel. To ensure the presence of sea ice, the OSI-450-a sea-ice concentration product is interpolated onto the initial parcel positions. Parcels with sea-ice concentrations below 15% are excluded from following processing steps.

### 2.2.2 Drift Correction

The drift correction is applied to parcels both forward and backward in time on a daily basis. The day onto which parcels are projected by applying the drift correction, is called *target day*. The OSI-455 displacements are first resampled onto the sea-ice concentration grid and then interpolated on parcel positions with a minimum sea-ice concentration of 15% using a bi-linear interpolation scheme.

The time bounds of the OSI-455 drift product span from 12:00 UTC on the previous day to 12:00 UTC on the day corresponding to the dataset registration time. The latter (12:00 UTC of the registration day) also serves as the reference time for the daily DA-SIT. Consequently, to correct drift on a given day, we use the drift product associated with a reference time set one day in advance.

For the initial drift correction of parcels, we first calculate the time difference between the registration time of the sea-ice parcel on the target day and the reference time of the drift product (12:00 UTC one day ahead). This difference is typically



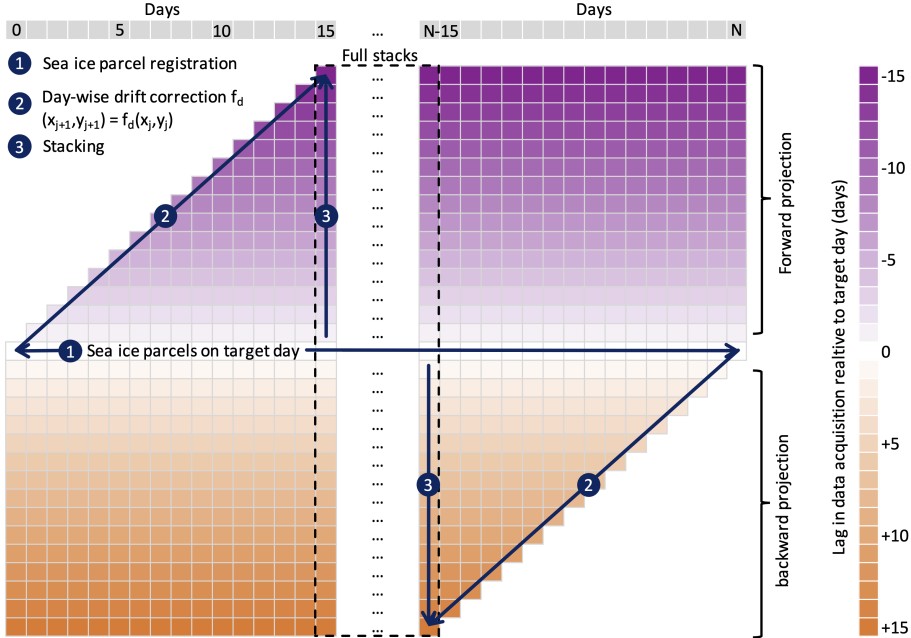

**Figure 3.** Processing scheme to produce the DA-SIT product. Each cell represents sea-ice observations of one day. The color map and vertical axis represent the delay regarding the time of data acquisition. Horizontal axis represents the ordinary timeline in day, e.g., day 0 may represent 1 October, and day N refers to 30 April. Diagonal aligned boxes (lower left to upper right for forward projection) represent daily trajectories, which are daily drift-corrected, containing the same SIT data from the day of acquisition.

close to 24 hours. Using this time difference, we adjust the position of the sea-ice parcel accordingly. Following this correction, the reference time for all parcels is standardized to 12:00 UTC of the respective day. Consequently, subsequent drift corrections involve a consistent 24-hour displacement.

Next, we use the OSI-450-a ice concentration corresponding to the day when the parcel reaches its updated position. Ice
concentration is interpolated onto these updated positions, and any parcels located in areas with ice concentration below a threshold of 15% are removed, as we assume that ice drifting into open ocean will melt even during polar winter. Therefore, those parcels are removed from the dataset. We also ensure that parcels do not unintentionally drift onto land grid cells, which can occasionally occur due to the low resolution of the drift product and related uncertainties.

As a result of the systematic drift correction, the original along-track SIT pattern disperses, transforming into a point cloud
where each point represents a circular parcel. This process is repeated daily, progressively updating the positions of the parcels over time. This dispersion is illustrated in Fig. 2. The map shows a selection of parcel trajectories with a length of 15 days. The parcels originate either 15 days prior to or 15 days after the target day. While at the time of origin, the parcels represent the satellite's track pattern, after 15 days, parcels have been drifted for partly more than 200 km.





### 2.2.3 Stacking Process

The stacking scheme is illustrated in Fig. 3. Processing typically begins on 1 October 1 (Day 0), the first day after the summer pausing when altimetry processing resumes. On this day, sea-ice parcels are initially registered following the procedure described in Section 2.2.1. The drift correction, as outlined in Section 2.2.2, is then applied to these parcels. Each elapsing day, newly registered parcels, or parcels already corrected for drift on previous days, are advected. For each parcel, a maximum of 15 daily drift corrections is allowed. Parcels are removed from the stack after the 15th correction.

This procedure is performed in both forward and backward time directions. Each day, drift-corrected parcels are stacked onto the drift-corrected parcels from one day prior or one day ahead, depending on the direction. A set of parcels is organized in a data frame structure, which contains the parcel geometries and their trajectories from the initial registration until the end of the drift (nominally 15 days).

The trajectory of each parcel is represented as a list of points, beginning with its initial registration position and ending at its final drift-corrected destination. Forward-projected parcels have trajectories progressing forward in time, while backward-projected parcels have trajectories moving backward in time.

As shown in Fig. 3, starting with a set of parcels on Day 0, the process produces a stack of 31 parcel sets by Day 15. This includes 15 sets from forward projections, 15 from backward projections, and the initial parcel set derived from the along-track SIT product on the target day. The diagonals in Fig. 3 represent the same parcels (and therefore the same SIT data) but at varying positions within the space-time domain due to daily forward or backward drift corrections. Ultimately, the vertically stacked parcel sets, spanning both past and future days, represent the predicted parcel positions on the target day.

After completing the forward and backward stacking, the parcel sets within a single stack are merged to create the drift-corrected dataset for a given target day (Fig. 3). Specifically, the parcel sets are combined into a unified data frame. Since both the forward and backward stacks include the parcel set corresponding to the target day, one of these is removed to prevent duplication.

### 2.2.4 Growth estimation and correction

Growth estimation aims to correct the SIT of parcels for processes that alter the thickness along their space-time trajectory. While parcels are drift-corrected, their SIT remains fixed at the state of initial registration. However, dynamic and thermodynamic processes alter the thickness between the registration time and the target day, which is typically up to 15 days. To account for these changes, we make use of multiple altimeter observations of approximately the same ice, facilitated by intersecting satellite orbits and drifting ice that leads to repeated measurements over the same parcels. The drift correction of all ice parcels ensures that when multiple parcels from different overflight times converge near the same position, they represent snapshots of the same ice parcel, but at different times.

It is important to note that, due to uncertainties in drift products and nonuniform altimeter measurement coverage within parcels, exact overlaps are unlikely. Nonetheless, we assume SIT values are representative within a given area, chosen here as $25 \times 25$ km, which is smaller than typical SIT correlation lengths in the Arctic (Ricker et al., 2017).



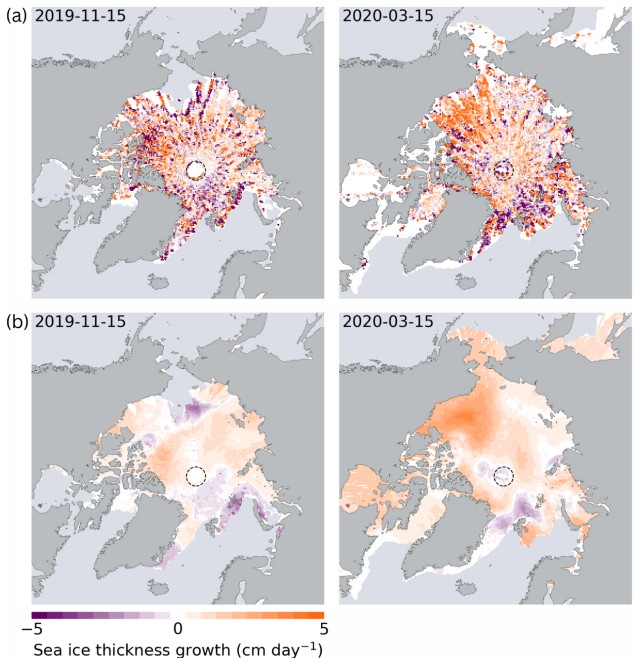

**Figure 4.** (a) Sea ice growth estimates from CryoSat-2 across the Arctic for November 2019 and March 2020 within a full stack of parcel sets covering one month. (b) Spatially interpolated sea ice growth based on growth estimates shown in (a). The white area in the background indicates data gaps with respect to the actual ice extent. The dashed circle represents the pole hole of the CryoSat-2 orbit coverage.

In the initial step of the growth correction process, we create a $25 \times 25$ km EASE2 grid. Next, we assign all parcels from the full stack to their corresponding grid cells. When a grid cell contains at least three parcels with SIT estimates from different points in time, we apply a least-squares linear fit:

$$f_H(n) = p_1 n + p_0 \tag{1}$$

where $f_H$ is the fitted sea ice thickness, and $n$ representing the number of days between the day of measurements and the target day. The coefficient $p_1$ represents the SIT growth, which can be positive or negative. It is important to note that, at this stage, we do not differentiate between thermodynamic and dynamic growth processes. Additionally, we acknowledge that this estimate is relatively coarse. The linear fitting also generates a covariance matrix, which quantifies the uncertainty of $p_1$. This uncertainty is later used for the overall uncertainty estimation. Fig. 4a illustrates an example of gridded growth estimates derived from a stack of trajectories centered on 15 November 2019 and 15 March 2020. These estimates represent ice growth over one-month periods. The density of growth estimates across the Arctic decreases with latitude as the orbit density decreases. Therefore, uncertainties of these estimates increase with distance from the pole.

To achieve a consistent growth correction, we fill the gaps that are left where not enough collocated parcels from different times could be identified. Here we use a radial basis function (RBF) interpolation in two dimensions with a Gaussian kernel based on the scipy.interpolate.RBFInterpolator function (Virtanen et al., 2020). An RBF interpolant based on data values $d$ at



locations $y$ is a linear combination of RBFs centered at $y$, and a polynomial $P(x)$ of a specific degree, evaluated at position $x$ (Virtanen et al., 2020):

$$f(x) = K(x,y)a + P(x)b \tag{2}$$

where $K(x,y)$ is an array of RBFs centered at $y$ and evaluated at $x$. The coefficients a and b solve the linear equations:

$$(K(y,y) + \lambda(\Phi)I)\,a + P(y)b = d \tag{3}$$

$$P(y)^T a = 0 \tag{4}$$

Here, $\lambda(\Phi)$ is a smoothing scale parameter, chosen to be latitude-dependent to account for the varying density of measurement points caused by the satellite's orbital geometry. The smoothing scale parameter decreases linearly from 80 at $\Phi = 40°\mathrm{N}$ to 10
at $\Phi = 90°\mathrm{N}$. For further details on the RBF interpolation method, we refer to Virtanen et al. (2020).

To minimize computation time, we limit the number of nearest data points used for each interpolation to a maximum of 260. Fig. 4b illustrates an example of spatially interpolated sea ice growth for 15 November 2019 and 15 March 2020, corresponding to Fig. 4a. The interpolated ice growth values are then applied to adjust the SIT estimates of the parcels, considering the number of days between the actual measurement date and the target day. Finally, parcel trajectories aggregated over one month are
exported in daily comma-separated value (csv) files (Fig. 3).

### 2.2.5 Computing performance

The computing efficiency of the stacking process benefits from parallel processing, which is implemented in the DriftAware-SIAlt python software. On a high performance computing cluster (HPC), using 16 cpu cores and 8 GB memory per cpu, processing of csv files with DA trajectories for an entire winter season (October-April) takes approximately 4 hours. A set of
daily csv files containing the DA trajectories for one winter season requires approximately 30 GB, with individual file sizes between 40 and 170 MB.

### 2.3 Evaluation on the EASE2 grid

We evaluate the stacked, now SIT growth-corrected, parcel sets on a 25 km EASE2 grid, in line with other existing SIT products (e.g., Ricker et al. (2017)) and the SIC climate data record. We average SIT of the parcel sets at their positions at
the respective target day within 25 km grid cells. This means we will retrieve daily DA-SIT grids, each containing data from a monthly window (15 days in both directions). Therefore, to provide gridded products that contain entirely independent data, we must consider the daily stacks with 1 month difference in reference time. To align with the conventional monthly gridded products, we select the mid-month stacks to compute the respective DA-SIT grids, i.e., on the 15th of each month. In any case we compute drift-aware products at any day during the winter season. This mean that during ramping phase at the beginning
and end of the processing period, stacks are not full, and therefore gridded maps will be incomplete within the first and last 15 days of the processing period (Fig. 3).

The average SIT ($H_{L3}$) for one grid cell is computed as an arithmetic mean, ignoring non-numeric values:



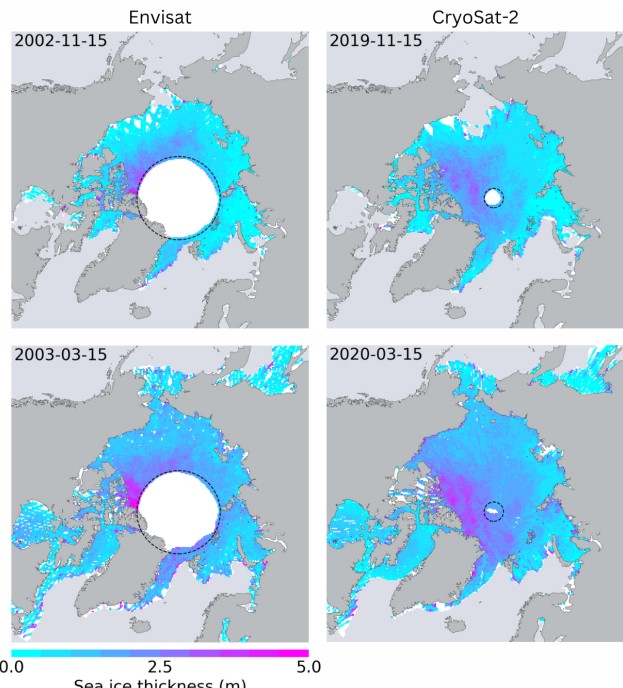

**Figure 5.** Drift-aware sea-ice thickness from Envisat (2002/2003) and CryoSat-2 (2019/2020) for November and March with applied sea ice growth correction. The white area in the background indicates data gaps with respect to the actual ice extent. The dashed circle represents the pole hole of the Envisat/CryoSat-2 orbit coverage.

$$H_{L3} = \frac{1}{n_{L2}} \cdot \sum_{i=0}^{n_{L2}} H_{i,L2} \quad \text{if } H_{i,L2} \neq \text{NaN} \tag{5}$$

where $n_{L2}$ is the number of parcels inside the corresponding grid cell.

Fig. 5 shows the final gridded DA-SIT for Envisat (November 2002 and March 2003) and CryoSat-2 (November 2019 and March 2020) after SIT growth correction. The drift-correction can cause parcels swapping into the pole hole, reducing its size. But the drift-awareness can also result in data gaps like in the Chukchi Sea in November 2002. Those gaps arise when the satellite's orbital drift aligns in both direction and magnitude with the drift of the sea ice. In fact, this means that DA-SIT also reveals sea ice areas that have never been surveyed by the satellite within the one-month observation period.

**2.4   Uncertainties**

We distinguish between three major contributors to uncertainties of the gridded DA-SIT:

1. The uncertainty associated with the along-track SIT retrieval, used as input for drift-aware processing, is derived from the Level-2 along-track SIT.





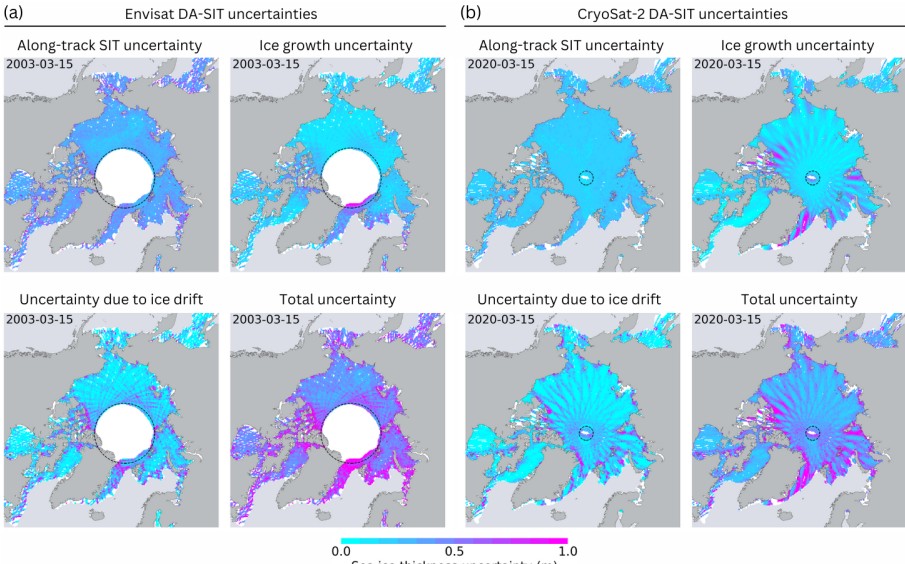

**Figure 6.** a) Uncertainty components of Envisat DA-SIT product for 15 March 2003. b) Uncertainty components of CryoSat-2 DA-SIT product for 15 March 2020. The white area in the background indicates data gaps with respect to the actual ice extent. The dashed circle represents the actual pole hole of the Envisat/CryoSat-2 orbit coverage.

The uncertainty of the mean SIT ($\sigma_{H_{parcel}}$) for each parcel is computed by considering the individual uncertainties of
measurements associated to the parcel:

$$\sigma_{H_{parcel}} = \frac{\sqrt{\sum_{i=1}^{n} \sigma_i^2}}{n} \qquad (6)$$

where $\sigma_i$ represents the uncertainty associated with the $i$-th measurement, and $n$ is the total number of measurements. This calculation assumes that the uncertainties are uncorrelated.

2. The uncertainty of SIT as a consequence of the drift uncertainty depends on two factors. First, the uncertainty of the
ice drift product, and second, the spatial variability of SIT in the area where the parcel might have been drifted into, considering the drift uncertainty. We start with aggregating the drift uncertainty along the parcel trajectory. In OSI-455, the daily drift uncertainty estimates, $\sigma_D$, are identical in both the $x$ and $y$ directions. For each step of drift correction, the squares of the drift uncertainties are aggregated. After the final drift correction, the drift uncertainty, $\sigma_{D_{x,y}}(t_j)$, of a parcel at the target day $t_j$ is computed as follows:

$$\sigma_{D_{x,y}}(t_j) = \sqrt{\sum_{i=j-N}^{j} \left( \sigma_{D_{x,y}}(t_i) \right)^2} \qquad (7)$$

where $N$ is the number of days over which drift correction is performed (up to 15 days), $t_j$ represents the time of the target day assigned the index $j$, and $i$ represents the day of data acquisition (the starting point of the drift). In the second





step, we analyze the impact of the spatial SIT variability. After the merging of forward and backward stacks including growth correction (Sections 2.2.3 and 2.2.4), all neighboring ice parcels within the area defined by the drift uncertainty are aggregated at each parcel's final position.. The aggregation uses a radius defined as $\sqrt{2}\sigma_{D_{x,y}}(t_j)$, where $\sigma_{D_{x,y}}(t_j)$ represents the estimated uncertainty in the $x$ and $y$ directions.

This results in a SIT distribution over the area where the ice parcel may have drifted, accounting for the drift uncertainty. If the thickness in this area is highly homogeneous, significant uncertainties in SIT are not expected, as the varying position of the ice parcel will not alter the SIT distribution within the area. Conversely, if the SIT in this area is highly heterogeneous, with substantial gradients, a different position of the ice parcel could result in changes to the SIT at its location. To estimate the potential uncertainty in SIT within the drift uncertainty radius, we calculate the inter-quartile range of the SIT distribution in that area, defined as the difference between the 75th and 25th percentiles. However, if the drift uncertainty radius does not exceed the radius of the parcel area, the resulting SIT uncertainty is set to 0. This is especially the case for fastened sea ice.

3. The uncertainty due to SIT growth and its correction is estimated from the covariance matrix obtained through linear fitting, which is used to estimate SIT growth (Section 3.6). For each parcel's growth estimate, the growth uncertainty, $\sigma_{H_G}$, is computed as follows:

$$\sigma_{H_G}(\Delta t) = \sqrt{\sigma_{p_1}^2}\Delta t \tag{8}$$

where $\sigma_{p_1}^2$ is the variance term of the growth coefficient $p_1$, derived from the diagonal elements of the covariance matrix. $p_1$ is multiplied by the time difference $\Delta t$, measured in days, between the target day and the day of data acquisition to calculate the SIT uncertainty due to ice growth for a given ice parcel.

The total uncertainty is calculated by the square root of the sum of the squares of the three individual components. Fig. 6 shows the three uncertainty components as well as the total uncertainty of DA-SIT for Envisat (15 March 2003) and CryoSat-2 (15 March 2020). The SIT uncertainty caused by ice drift is significantly smaller in the pack ice but becomes more pronounced in the marginal ice zone and especially along coastlines. This is primarily due to the greater uncertainty in drift products near coastlines, whereas the relatively low uncertainties in the pack ice zone often result in accumulated drift uncertainties that remain smaller than the radius of the parcel cell. It is also worth noting that examples from recent years benefit from the AMSR2 mission, starting in 2012. Earlier years, dominated by the SSM/I missions, may exhibit larger drift-related uncertainties.

The SIT uncertainty due to ice growth shows a distinct orbital pattern, a result of the growth uncertainty being a function of the elapsed time between data acquisition and target day.





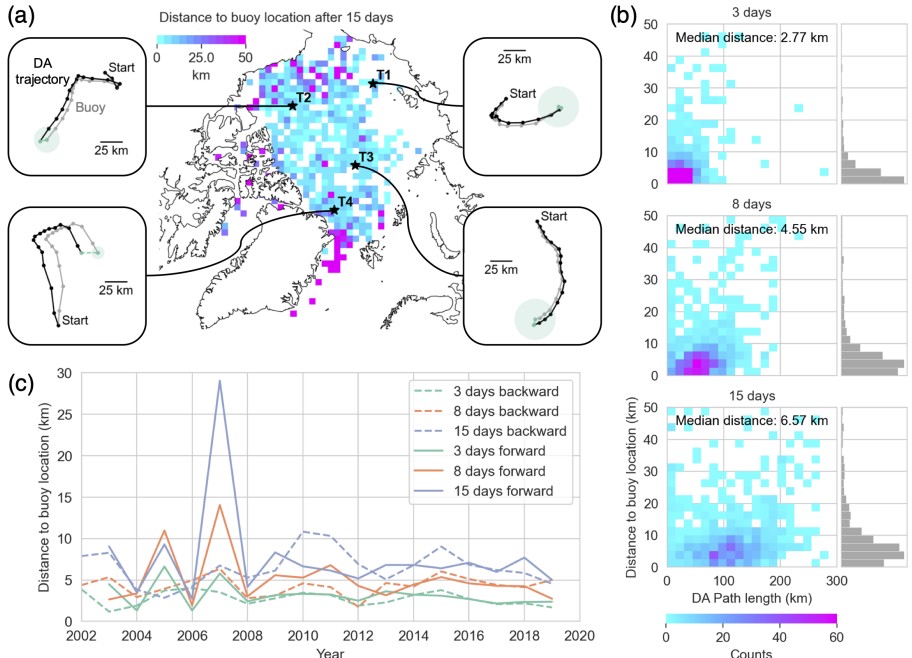

**Figure 7.** (a) Validation of DA trajectories with buoys, including example trajectories (T1-4). Green-filled circles represent the DA-uncertainty of the drift correction after ±15 days. (b) Distance between DA-SIT and buoy trajectory after 3, 8 and 15 days, as a function of DA path length. (c) Yearly mean distance between DA and buoy trajectory after 3, 8 and 15 days in both directions.

## 3 Validation of the Drift-Awareness Sea Ice Thickness

### 3.1 Validation of DA-SIT Trajectories using Buoys

To validate the DA-SIT trajectories, we use buoy data provided as part of the International Arctic Buoy Program (IABP). The IABP includes different types of buoys that measure snow and ice parameters such as ice temperature, snow depth, and SIT. However, for this study, we only use the buoy positions. Although IABP also includes ice mass balance buoys to measure sea ice thickness, we do not use them for the SIT evaluation, as their measurements only represent the sea ice at their immediate position, and are therefore not suited for comparing with satellite observations, integrating sea ice over at least several hundred meters. In total we use 1029 buoys. The original sampling frequency of the buoy data was 3-hourly until 2016 and hourly thereafter. We resample the IABP buoy data to daily positions at 12:00 UTC to align with the reference time of the DA-SIT trajectories. The daily buoy positions are then divided into 31-day periods, centered on a target day with a ±15-day window to align with the DA-SIT trajectories. For the starting dates of the 15-days buoy trajectories, we identify the spatially closest initial DA-SIT ice parcels from all available trajectories. This includes both trajectories in forward and backward direction. We set a maximum distance threshold of 25 km between the buoy and the initial DA-SIT ice parcel position. This threshold is



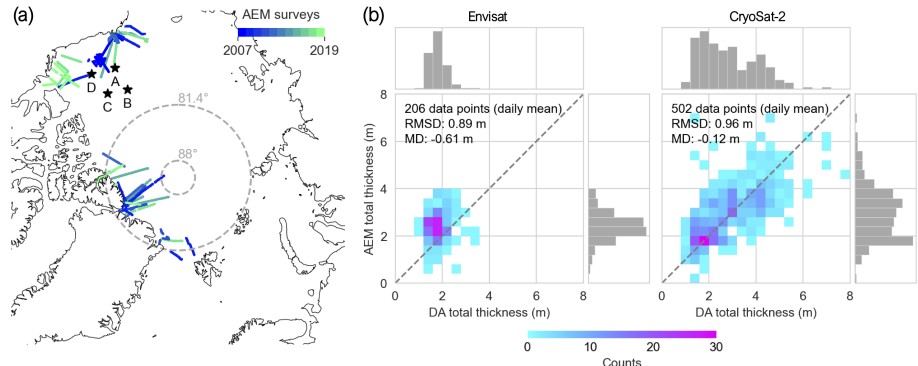

**Figure 8.** (a) Overview of DA-SIT validation data across the Arctic. AEM surveys are sorted by year. The black stars indicate the locations of the BGEP moorings. (b) DA-SIT validation heatmaps, separated between the Envisat and CryoSat-2 eras.

well below the spatial resolution of the drift product, minimizing any significant impact of positional offsets between the initial positions of the parcel and the buoys.

Fig. 7a shows a map of distances between the end points of individual DA-SIT trajectories, and the corresponding buoy trajectories. We consider only the distances after 15 days of drift, either backward or forward in time. The pattern indicates higher uncertainties of the DA-SIT trajectories in the Fram Strait and East Greenland Sea as well as in the marginal ice zones and in peripheral locations. Here the discrepancy after 15 days of drift correction can reach more than 50 km. In contrast, the Central Arctic shows distances of typically less than 10 km, which is below grid-cell size.

Fig. 7b shows the linkage between the uncertainty of the DA-SIT trajectory and path length. The path length is the covered distance along the trajectory. It begins on the day of the measurements and ends on the target day and is generally increasing with time. With increasing time lag and path length, also the distance between buoy location and DA ice parcel location is increasing. While after 3 days, we observe a median distance of 2.77 km considering all co-registered trajectories, it increases to 6.57 km after 15 days. Therefore, we conclude that with increasing DA time lag, the uncertainty in the respective ice parcel location increases. This uncertainty is described in section 2.4 and displayed in the four trajectory examples in Fig. 7a. Fig. 7c shows the yearly mean distance between DA-SIT parcel and buoy location for all co-registered trajectories across the Arctic, after 3, 8, and 15 days. There is no significant trend in the distances between buoy and DA ice parcel within the 2002-2020 period. We also do not observe a significant difference in the distances between backward and forward projected trajectories. However, as discussed before (Fig. 7b), we observe a dependency of the distance on the time lag. The longer the time lag between target day and day of measurements, the higher the discrepancy between buoy and DA-SIT parcel location.

### 3.2 Validation with Airborne EM

Airborne electromagnetic induction thickness sounding (AEM) is a method to measure total sea ice thickness directly. The EM-Bird is an AEM sensor that is towed over sea ice by a helicopter or fixed-wing aircraft (Pfaffling et al., 2007; Haas et al., 2009). The distance to the sea ice/seawater interface is calculated from the EM response. The sea ice + snow/air interface is



obtained from a laser altimeter. The difference between the two interfaces corresponds to the sea-ice + snow thickness. The point spacing of the measurements is about 5 m.

We use AEM data from helicopter and fixed-wing aircraft campaigns during the period 2007-2019 (Table A1). AEM data are projected and averaged on a 25 km EASE2 grid, in line with the gridded DA-SIT. As the AEM data represent total thickness, we
have to convert the DA-SIT into total thickness. In the drift-aware processing, the snow depth inherited from the L2 along-track data is assigned to parcels in the same way as SIT, and is therefore also drift-corrected. Therefore, we add the gridded snow depth to the DA-SIT retrieval to retrieve DA total thickness. Daily grids of AEM data are then compared with the matching DA-SIT grids. The validation of the gridded DA-SIT product against total ice thickness is carried out for both the Envisat and the CryoSat-2 era and allows to evaluate the spatial distribution of SIT at a given time. Due to the larger Envisat pole hole
(>81.4°N), suitable AEM surveys are primarily limited to the Beaufort and Chukchi Sea (Fig. 8a). Therefore, the surveyed ice does not contain as thick and deformed ice as in the Lincoln Sea for example. The CryoSat-2 orbit domain goes up to 88°N and therefore also includes AEM surveys over thick, deformed multiyear sea ice. Fig. 8b shows validation results for each satellite era. The mean difference (MD) is -0.61 m for Envisat and -0.13 m for CryoSat-2. The slight underestimation of thickness observed in the Envisat data is also evident in the CCI CDR dataset, attributed to the differences in altimeter types,
specifically pulse-limited versus Doppler-delay (Paul et al., 2018). The root mean square deviation (RMSD) is quite similar for both sensors, 0.89 m for Envisat and 0.98 m for CryoSat-2.

### 3.3 Validation with ULS

Upward looking sonars (ULS) are mounted on oceanographic moorings (Hansen et al., 2013; Krishfield et al., 2014) and provide information on long-term ice thickness variability and seasonal changes. ULS are used to derive ice draft by measuring
the travel time of a sonar pulse transmitted by the ULS and reflected back from the ice bottom. The ice draft can then be converted into ice thickness assuming snow load and densities of snow and ice. In contrast to AEM, ULS are valuable to evaluate the temporal change in sea ice draft in the vicinity of the ULS location. We use ULS ice draft data from moorings that have been deployed at four different sites in the Beaufort Sea within the Beaufort Gyre Exploration Project (BGEP, Krishfield et al. (2014)). Fig. 8a shows the position of the moorings in the Beaufort Sea. Their draft time series cover the Envisat (A, B,
C, D), as well as the CryoSat-2 era (A, B, D). Exact locations and data record periods for the ULS A-D are provied in Table A2.

The original data are sampled at 2s intervals. We first remove open water sections. Afterwards, the filtered data were averaged over 24 h to obtain daily mean effective ice thickness for each ULS. Then, for each day and each ULS position, we co-register the nearest three grid cells in the DA-SIT grids. The thicknesses and uncertainties of the three grid cells are averaged. As
the ULS provide draft estimates, we convert the co-registered DA-SIT into ice draft using climatological densities for ice $(900\,\mathrm{kg\,m^{-3}})$, snow $(300\,\mathrm{kg\,m^{-3}})$, and water $(1025\,\mathrm{kg\,m^{-3}})$. The snow depth is obtained from the DA-SIT product, which contains snow depth inherited from the L2P along track data.

Fig. 9 shows results from the comparison between the ULS and sea ice draft derived from DA-SIT (DA draft). For reference, the figure also includes ice draft derived from conventional gridding, which does not apply drift-awareness. This means that



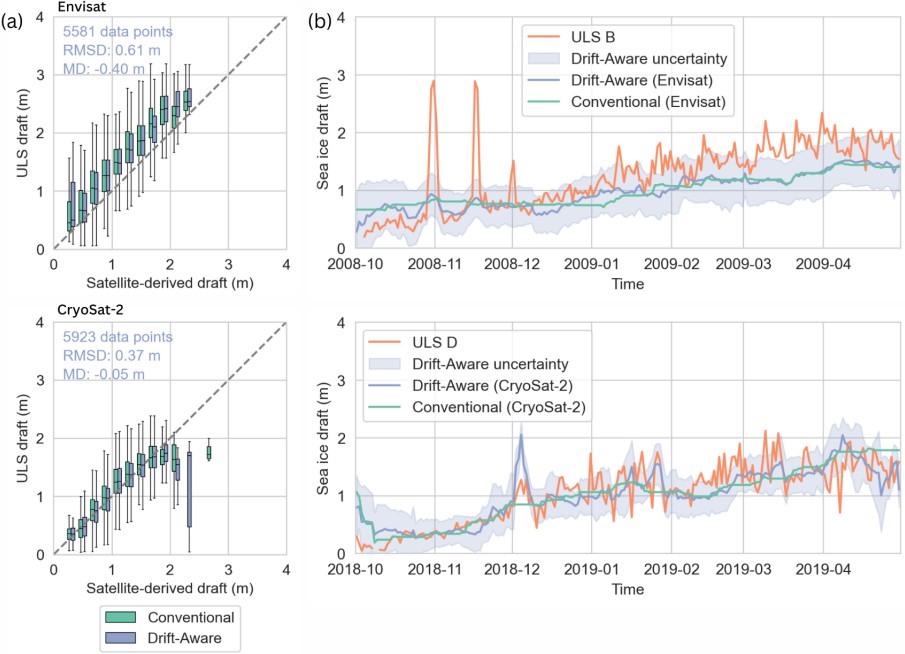

**Figure 9.** (a) Box and Whisker plots of daily DA draft and ice draft from conventionally gridded SIT from Envisat (2002-2012) and CryoSat-2 (2010-2020) against daily averaged ULS draft. (b) Daily sea ice draft from ULS and the co-registered DA draft as well as ice draft from conventionally gridded SIT over the winter seasons 2008/2009 (CryoSat-2) and 2018/2019 (CryoSat-2).

along-track SIT retrievals are gridded daily, aggregating data within a moving time interval of one moth. The comparison with the conventionally gridded data is discussed in Section 4. Fig. 9a shows the comparison between all BGEP ULS and DA draft for the Envisat (2002-2012) and CryoSat-2 (2010-2020) eras. Combining all ULS, we find a mean difference (MD) of -0.4 (-0.05) m and a root mean square deviation (RMSD) of 0.61 (0.37) m for the comparison with Envisat (CryoSat-2) DA-SIT. Considering Envisat DA-SIT, results suggest that SIT is generally underestimated, agreeing with the results from the AEM

comparison. Again, this is also in agreement with the findings of the validation of the CCI SIT CDR. Khvorostovsky et al. (2020) provide a detailed comparison between BGEP ULS data and the CCI SIT CDR. They conclude that the sea ice draft growth underestimation observed for the most of winter seasons depends on the surface properties preconditioned by the melt intensity during the preceding summer. In our study, this is particularly true for Envisat, while CryoSat-2 DA draft shows a better agreement with regard to the RMSD and MD. This is expected because of the smaller footprint of CryoSat-2, and

therefore higher sensitivity for deformed, thick ice. The discrepancy for draft classes > 2m for CryoSat-2 DA draft may result from the small sample size (< 10) per draft class.

While the Box and Whisker plots for all ULS in Fig. 9a provide a general assessment, they do not capture the actual co-variability between datasets. To address this, we also evaluate the seasonal evolution of co-variability between the daily ice





draft measured by the ULS and the co-registered DA-derived draft. Fig. 9b shows two winter seasons: 2008/2009 with ULS B
ice draft compared to Envisat DA-draft, and 2018/2019 with ULS D ice draft compared to CryoSat-2 DA-draft.

The DA draft captures the overall thermodynamic growth signal throughout the winter. In contrast, the short-term variability
observed in the ULS data arises from changes in the ice draft as ice floes of different types and thickness drift through the
sonar beam. This variability is driven by sea ice dynamics, where thicker, deformed ice and ridges formed during convergence
coexist with thinner, newly formed ice resulting from divergence. The drift-awareness algorithm provides colocation on a
spatial scale of a few kilometers, as discussed in Section 3.1. Moreover, the DA draft is affected by uncertainties originating
from the processing of the altimetry raw data (Ricker et al., 2014). Consequently, the ice draft measured by the ULS and the
co-registered DA ice draft often do not align precisely. As a result, we presume that the variability in DA and ULS draft is
frequently out of phase.

However, some periods are in good agreement between the two datasets. For instance, in autumn 2008 we observe two
distinct anomalies in the ULS B draft that can be also observed in the DA draft (Envisat), although weaker. Similar, at the
beginning of December 2018, both ULS D and DA draft (CryoSat-2) reveal a sharp increase in ice draft, likely caused by
a cluster of very thick ice drifting over the ULS. We believe that due to the limited resolution of the drift product and the
differences in the illuminated areas between satellites and ULS, even after applying the DA algorithm, daily SIT averages of
ULS and parcel thickness will be out of phase. But Fig. 9b proves that in some cases, the DA-algorithm captures SIT anomalies
observed by the ULS.

## 4 Impact Analysis

### 4.1 Time and Distance Offset to Data Acquisition

For each parcel, we track the time offset between the start and end of each parcel trajectory. The maximum offset is given
by the time widow, that defines over how many days along track data are drift-corrected and stacked. In this study, we use ±
15 days to be in line with the satellites sub cycle. Fig. 10a shows the gridded time offset for Envisat and CryoSat-2 DA-SIT
parcels respectively, considering the beginning (November) and end (April) of the winter season. Due to the drifting satellite
orbit, we observe a track pattern, with alternating time offsets in latitudinal direction. The thickness change along the parcel
trajectory correlates with the time offset, because of thermodynamic ice growth as well as the accumulation of deformation. As
a consequence, neglecting ice drift and ice growth corrections, the time offset pattern will cause track patterns in the monthly
SIT maps.

Similar, we calculate the geodesic distance between the final parcel position after drift correction, and the position at the
time of the measurements. Fig. 10b shows the gridded geodesic distance for the same sensors and months as in Fig. 10a. The
distance depends on mainly three factors. First, the travel distance scales with the temporal offset (Fig. 10a) for a given sea ice
motion vector. Second, the magnitude of the sea ice drift, and third, it's direction affects how far away from the location of
the satellite overflight the parcel has been traveled. For example, high, southerly directed ice drift in the Fram Strait leads to
large distances to the location of data acquisition, scaled by the time offset. Those patterns are highly variable as they directly





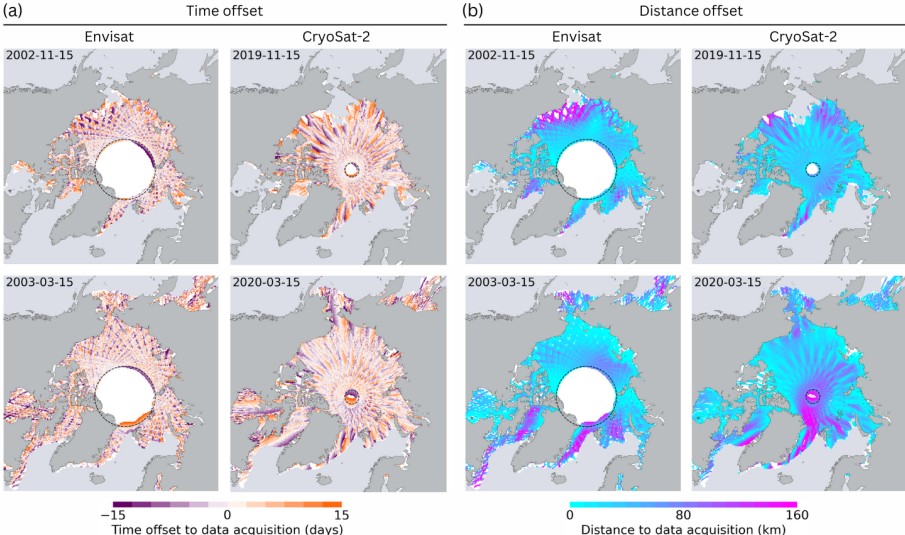

**Figure 10.** (a) Gridded time difference between the target day and the data acquisition time for Envisat (2002/2003) and CryoSat-2 (2019/2020), for 15 November and 15 March. (b) Same as (a) but for the geodesic distance between parcel position on the target day and at the time of the data acquisition. The white area in the background indicates data gaps with respect to the actual ice extent. The dashed circle represents the pole hole of the Envisat/CryoSat-2 orbit coverage.

depend on the sea ice drift. In the Fram Strait in March 2020, we find distances of up to 200 km, which is equal to the width of 8 grid cells. This means, neglecting the ice drift correction in regions of high drift rates, will cause incorrect localization of thickness anomalies in heterogeneous sea ice regimes.

## 4.2 Comparison with conventional Sea Ice Thickness Grids

The key difference between the DA-SIT product and the baseline CCI SIT is the drift-awareness as DA-SIT is based on the same along-track retrievals. Therefore, we compare DA-SIT with the conventionally gridded sea ice thickness (C-SIT). This means that we use the parcel positions at the time of the satellite overflight. Here we only present the results from the CryoSat-2 era, as we find a very similar results for the Envisat era. Fig. 11a shows the difference map between DA-SIT and C-SIT for 15 March 2020. We divide the Arctic into maritime regions, introduced by Meier and Stewart (2023). For these regions, we calculate the mean difference and standard deviation of differences between DA-SIT and C-SIT, shown in Figure 11b). The example from 15 March 2020 highlights regional differences in the magnitude of the impact of drift-awareness. In the East Greenland Sea, including Fram Strait, we generally find the highest standard deviation in differences, while the lowest standard deviation in differences for March 2020 is found in the East Siberian Sea. Considering the entire CryoSat-2 era (2010-2020), the Arctic Ocean exhibits the lowest standard deviation. In fact, the results for March 2020 appear to be representative overall. The mean differences in selected regions can be either positive or negative. As displayed by the histograms of differences in Fig. 11a, the distributions can be close to Gaussian shape (e.g. East Siberian Sea), but can also be slightly skewed (e.g., East Greenland Sea).





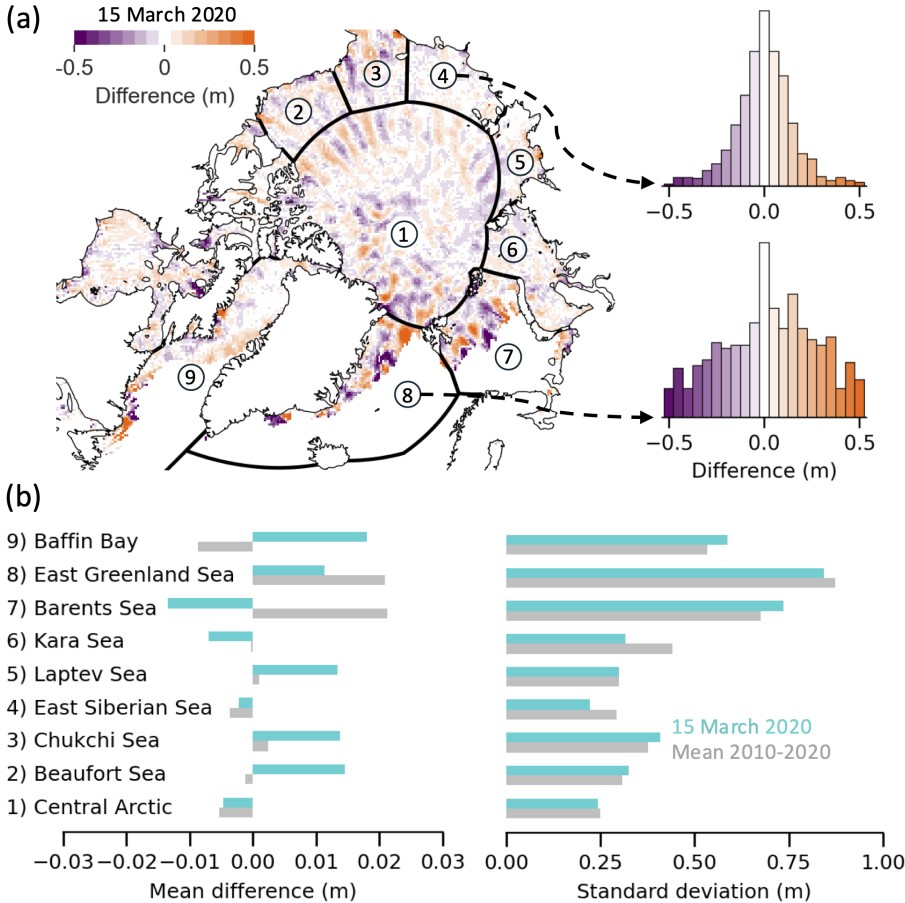

**Figure 11.** Intercomparison between DA-SIT and conventionally gridded SIT. (a) Difference map for 15 March 2020, including histograms of differences for East Siberian Sea and East Greenland Sea. (b) Mean difference and standard deviation for marine regions for 15 March 2020 and the mean over the 2010-2020 period.

Typically, the smaller the considered area is, the higher the mean differences. The pattern of differences in Fig. 11a reveals that the strongest impact of drift-awareness is found in regions with strong sea ice drift, e.g., Beaufort Gyre and Fram Strait. In the Beaufort Sea region and north of it, we observe an alternating pattern of positive and negative differences in longitudinal direction. This originates from the fact that the C-SIT does not take into account the drifting along track measurements, which contributes to the typical orbit track pattern (known as trackiness), which sometimes can be observed in the conventional SIT maps (Ricker et al., 2014). As expected, the DA-SIT product reduces this effect. The overall mean difference between DA-SIT and C-SIT is 0.29 cm for the period 2010–2020 across all regions, which means that the impact on mean Arctic SIT and volume time series is negligible.



## 5 Conclusions

We have presented a method to ensure drift-awareness (DA) for sea ice thickness (SIT) maps derived from satellite altimetry and demonstrated its application on Envisat and CryoSat-2 data within the framework of the ESA Climate Change Initiative. The study included method descriptions, validation, and impact analysis. By applying DA, SIT retrievals from one month of altimetry measurements are projected onto one day, allowing to produce daily maps of Pan-Arctic SIT. This approach facilitates comparisons with SIT derived from passive microwave radiometer measurements (Tian-Kunze et al., 2014) or in-situ observations. From our findings in this study, we draw the following conclusions:

1. **Validation with Observational Data:** The DA-SIT method can capture anomalies observed in daily upward-looking sonar (ULS) averages, depending on the spatial extent of the anomalies and accurate colocation of measurements. This highlights the method's capacity for improved regional analyses, though uncertainties in drift corrections must be considered.

2. **Improvement in spatial representation:** Drift-awareness improves the spatial representation of sea ice thickness in gridded products by reducing regional biases (10–20 cm) and minimizing displacement errors (up to 200 km) caused by neglecting sea ice drift. This improvement is particularly important in regions with strong drift, such as the Beaufort Sea and East Greenland Sea.

3. **Regional vs. Pan-Arctic Effects:** While regional-scale biases can be significant, the pan-Arctic mean differences between drift-aware SIT (DA-SIT) and conventional SIT maps are negligible (0.29 cm across all regions from 2010–2020, Fig. 11). This is because regional biases balance out on a larger scale, as ice is redistributed rather than lost.

4. **Applications across missions:** The DA algorithm is applicable to other satellite altimeters such as ERS-1/2, Sentinel-3A/B, and ICESat-2. Drift-awareness should be integrated into future SIT mapping efforts to reduce uncertainties, especially when merging data from different missions, for example radar (e.g., CryoSat-2) and laser altimeters (e.g., ICESat-2) to derive snow depth.

5. **Future altimetry missions:** Drift-awareness will be particularly important for new advanced altimeters, such as the Copernicus Polar Ice and Snow Topography Altimeter (CRISTAL) (Kern et al., 2020). To fully leverage the high-resolution measurements these systems provide, DA should be combined with higher-resolution drift products, such as synthetic aperture radar (SAR) data (Howell et al., 2022) or future radiometer observations like the Copernicus Imaging Microwave Radiometer (CIMR) (Lavergne et al., 2021).

Overall, incorporating drift-awareness in sea ice thickness mapping enhances the accuracy of gridded products, supports climate monitoring, and improves the understanding of sea ice dynamics. As an outlook, we plan to apply the drift-awareness on altimetry retrievals in the Southern Ocean. Moreover, implementation of a column model that estimates dynamic and thermodynamic SIT growth along the DA trajectories could provide complementing information, which help to interpret the satellite-derived SIT estimates.



*Code and data availability.*  All data that have been used or produced in this study are publicly available:

– The DA-SIT data record (v100) is available on Zenodo: https://doi.org/10.5281/zenodo.14733131, last access 24 Jan 2025.

– Sea ice drift (OSI-455) and concentration (OSI-450-a) were obtained from ftp://osisaf.met.no, last access 20 Jan 2025.

– Envisat and CryoSat-2 level-2 along track data, produced within the ESA CCI project, were obtained from the ftp server hosted by the Alfred Wegener Institute (ftp.awi.de/sea_ice/projects/cci/crdp/v3p0, last access: 20 Jan 2025).

– AEM data were obtained from the PANGAEA data portal (Felden et al., 2023) (https://www.pangaea.de, last access: 20 Jan 2025).

– BGEP ULS data were obtained from the Woods Hole Oceanographic Institution (https://www2.whoi.edu/site/beaufortgyre/data/, last
access: 20 Jan 2025).

– Bouy data were obtained from the IABP (https://iabp.apl.washington.edu/data.html, last access: 20 Jan 2025):

– 3-hourly data (until 2016) were obtained from https://iabp.apl.washington.edu/Data_Products/BUOY_DATA/3HOURLY_DATA/, last access: 20 Jan 2025.

– Hourly data (after 2016) were obtained from https://iabp.apl.uw.edu/WebData/LEVEL2/, last access: 20 Jan 2025.

A snapshot of the Drift-Aware python toolbox used to produce the results in this study is available on
Zenodo: https://doi.org/10.5281/zenodo.14732875, last access 24 Jan 2025.

*Video supplement.*  The video supplement contains an animated timeseries of DA-SIT maps from 2019–2020.

## Appendix A:  Validation Datasets Overview

### A1   AEM datasets

Table A1 lists the AEM data records for the period 2007-2019, which are used for validation in this study as well as the surveyed regions and used platform.

### A2   ULS datasets

Table A2 provides the position of the moorings in the Beaufort Sea and information about the ULS data record periods. They cover the Envisat (A, B, C, D), as well as the CryoSat-2 era (A, B, D).

*Author contributions.*  Original idea of the Drift-Aware approach: RR, TL Design and development of the Drift-Aware algorithm: RR, MB. Sea ice drift data input and interpretation: TL, ED. Sea ice thickness data input and interpretation: SH, SP. Project support: MAK. Discussion of results and conclusions: All. Writing the manuscript: All.

*Competing interests.*  The authors declare that they have no conflict of interest.



*Acknowledgements.* RR and MB were supported by ESA CCI Sea Ice (CCN-2 to Contract 4000126449/19/I-NB -Sea_Ice_cci), the Fram
Centre project "Sustainable Development of the Arctic Ocean" (SUDARCO) (project_ID: 2551323), and the Research Council of Norway
project "Thickness of Arctic sea ice Reconstructed by Data assimilation and artificial Intelligence Seamlessly" (TARDIS) (grant 325241).
TL and MAK were supported by ESA CCI Sea Ice (CCN-2 to Contract 4000126449/19/I-NB -Sea_Ice_cci). ED was supported by the
EUMETSAT OSI SAF fourth Continuous Development and Operations Phase (CDOP4).





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





**Table A1.** AEM data record overview used in this study. Data are only considered if they match with the respective satellite product coverage.

| Campaign Name | Period | Region | Aircraft Type |
|---|---|---|---|
| SEDNA | 05.04.2007 - 13.04.2007 | Beaufort Sea/ Chukchi Sea | Helicopter EM |
| SIZONet | 12.04.2008 - 14.04.2008 | Beaufort Sea/ Chukchi Sea | Helicopter EM |
| PAMARCMIP | 05.04.2009 - 26.04.2009 | Beaufort Sea/ Chukchi Sea/ Fram Strait | Fixed-wing aircraft EM |
| SIZONet | 09.04.2010 - 12.04.2010 | Beaufort Sea/ Chukchi Sea | Helicopter EM |
| PAMARCMIP | 31.03.2011 - 28.04.2011 | Beaufort Sea/ Chukchi Sea/ Lincoln Sea/ Fram Strait | Fixed-wing aircraft EM |
| PAMARCMIP | 03.04.2012 - 05.04.2012 | Beaufort Sea/ Chukchi Sea/ Lincoln Sea | Fixed-wing aircraft EM |
| SIZONet | 07.04.2012 - 09.04.2012 | Beaufort Sea | Helicopter EM |
| SIZONet | 30.03.2013 - 03.04.2013 | Beaufort Sea | Helicopter EM |
| SIZONet | 04.04.2014 - 05.04.2014 | Beaufort Sea/ Chukchi Sea | Helicopter EM |
| PAMARCMIP | 07.04.2015 - 23.04.2015 | Beaufort Sea/ Chukchi Sea/ Lincoln Sea | Fixed-wing aircraft EM |
| PAMARCMIP | 21.03.2017 - 08.04.2017 | Arctic Ocean/ Fram Strait/ Beaufort Sea/ Chukchi Sea/ Lincoln Sea | Fixed-wing aircraft EM |
| ICEBIRD | 01.04.2019 - 10.04.2019 | Beaufort Sea/ Chukchi Sea/ Lincoln Sea | Fixed-wing aircraft EM |

**Table A2.** Mooring sites with ULS measurement periods used in this study.

| Mooring Site | ULS Record Periods | Location |
|---|---|---|
| A | 08/2003 – 04/2020 | 150.0°W 75.0°N |
| B | 08/2003 – 09/2009 | 150.0°W 80.0°N |
|   | 10/2010 – 04/2020 |   |
| C | 08/2003 – 07/2008 | 140.0°W 77.0°N |
| D | 09/2006 – 04/2020 | 140.0°W 74.0°N |