# Peer review of "Drift-aware sea ice thickness maps from satellite remote sensing"

_EGUsphere, 2025_

## Referee Comment (RC1)

The manuscript "Drift-aware sea ice thickness maps from satellite remote sensing" by Ricker et al. describes a new algorithm for aggregation of along-track sea ice thickness measurements using corrections from satellite-derived sea ice drift. The produced sea ice thickness dataset is thoroughly validated and supplied with an uncertainty estimate. The manuscript presents new results that are important for climate and cryosphere research. It is well structured and contains all the necessary algorithm details and dataset description details. Nevertheless, a few open questions need to be addressed in the manuscript. I believe it can be recommended for publication after a major revision.

**Major comments**

The text contains grammar mistakes, and a thorough check by a native English speaker is recommended. I marked as much as I could, but I'm not a native English speaker either.

It is good that a separate validation of ice drift trajectories used for tracking the parcels is included. However, the presented algorithm (a data-driven model) contains one more module that can potentially significantly affect thickness distribution: accounting for thermal growth. How accurate is that one alone? It seems important to show the RMSEs computed between the input SIT in the parcels (Line 163), and the fitted values ($f_H$, Line 165). It will probably be proportional to the presented growth uncertainty, but RMSE is a more robust method.

The authors used arithmetic mean (Line 207) to compute SIT from advected parcels. Given that the uncertainty of SIT increases with the advection time (both due to growth and drift), would it be more reasonable to use a weighted average with weights inversely proportional to the time of advection? Alternatively, a limit on trajectory length can be added to prevent points from drifting too far (see also comment below on thick ice blocks in MIZ).

It is not clear from the description how the ice drift is used for advecting parcels, i.e., how the ice velocity is integrated to obtain displacement? Is it a linear Euler method or a more accurate Runge-Kutta 4th order, typically used for computing trajectories in inhomogeneous flows? What is the impact on the accuracy of the advection of the selected method (especially in areas with strong ice drift, e.g., Fram Strait).

Validation of a non-DA-SIT should be included in section 3.2 Validation with Airborne EM, to be consistent with Section 3.3 Validation with ULS, and illustrate the impact of data awareness. Expectedly, the impact will not be significant, but it needs to be documented.

It would be beneficial to homogenize presentation of the comparison – use either 2D histogram (like in Sec. 3.2), or a scatterplot with binning (like in Sec. 3.3). Additionally, the reference data (the independent variable) should be placed on X-axis, and the tested data – on Y-axis, following a common practice.

It is important to add Pearson's correlation or coefficient of determination to the provided RMSE and bias. It's expected to be quite high for the CS2 DA-SIT vs AEM-sit, for example, but nearly zero for Envisat, indicating that it may have low predicting skills for SIT over 2 m.

One of the applications of satellite-derived SIT is operational monitoring or assimilation into forecasting models. How efficient can the proposed algorithm be for such an application? Would advection over the last 15 days be sufficient (and important) for providing a more accurate SIT distribution in NRT? Can it be illustrated in the manuscript?

On the video supplement, in mid-January, I noticed some extremely high thicknesses in the MIZ in the Greenland Sea. What is the source of that? It should be either fixed or well presented in the paper with explanations.

[Figure]

Figure 4. a, left.

[Figure]

There are lines with alternating very high and very low growth rates along the CS2 orbits. Is that an error in the estimation of the growth coefficients or divergence/convergence of sea ice? More light needs to be shed on these cases (e.g., actual time series of advected SIT, divergence estimate from drift fields). It influences a lot the RBF-gridded SIT growth fields at a later stage but doesn't look as convincingly accurate data.

**Specific comments**

Line 2. Operational products are averaged over 2 weeks period, "one-month period" need to be rephrased.

Line 5. The acronym DA is very often used for Data Assimilation. May authors consider some other abbreviation for the product name to avoid confusing the users who have not read the paper. "Advected SIT" maybe?

 Line 10. "enables" -> "facilitates"

Line 15. "to a state more closely to" -> "to a state similar to"

Line 17. "on a large scale"

Line 26. "one month" is not a standard for, e.g., operational products.

Line 41. "might get lost" -> "is neglected"

Line 44. "observer" -> "observe"

Line 45. Either "single-synchronized orbits" or "a single synchronized orbit"

Line 98. Improve sentence consistency, e.g.: "Location of parcel centres is selected using the NSIDC EASE-Grid 2.0 (Brodzik et al., 2012) with spacing of10 km in both x and y directions."

Line 108. What is the uncertainty of the interpolation? Can it be added to the total uncertainty budget in Sec. 2.4?

Line 116. It is not clear how the position is adjusted. Please clarify in the text.

Line 117. "Consequently, subsequent" -> "Subsequent"

Line 128. -> "some parcels have drifted for more than 200 km."

Line 193. "Python"

Line 211. "swapping" -> "sliding"

Line 223. How reasonable is the assumption that the uncertainties are uncorrelated? In principle, the same (or very similar) sea ice is sampled by an altimeter in one parcel. With the same roughness that largely contributes to the uncertainty.

Line 225. Is the uncertainty of the interpolation procedure taken into account?

Line 261. Previously the product was called "drift-aware SIT", which sounds better than "Drift-Awareness SIT" (although, as mentioned earlier a better name can be found to

avoid confusion with data assimilation abbreviation). Please check that the product and the algorithm are called consistently.

Line 341. The sample sizes for all bins on Fig. 9 should be provided.

Line 359. "Proves" seems to be an overstatement. Fig. 9b indeed illustrates that in some cases, there is a coincident increase of thickness in both products. However, it also shows that in some cases, it is the opposite (e.g., 10 – 20 Dec 2008). For proving, it is crucial to include r, or r2 scores for the comparison of ULS and DA-SIT, as mentioned earlier.

Line 367. "latitudinal" -> "meridional"

Line 373. "on mainly three factors" -> "mainly on three factors"

Line 375. ""

Line 383. It is not very clear. Does the conventional product use "parcel positions at the time of the satellite overflight"? Please rewrite.

Line 386. Quite often "mean difference" is called bias, and "standard deviation of differences" is called RMSD (or RMSE). I would suggest using these terms here and below.

Line 390. I would not agree with "In fact, the results for March 2020 appear to be representative overall.", especially for the biases. Maybe this sentence can be simply excluded? As well as the next one, which doesn't bring any useful information.

Line 392. The peak on Fig. 1.a looks quite sharp, more typical for a Laplace distribution. Is the actual type of the distribution important or just the fact that it is symmetrical? Maybe "Gaussian" can be avoided?

Figure 11. Please add bias and RMSE for the entire Arctic.

Line 393. ", the higher the mean differences." -> "", the higher are the mean differences.", or rather ", the higher are the biases."

Line 394. Do you mean the Barents Sea instead of the Beaufort Sea? The Barents Sea has the second largest bias, and the mean bias in the Beaufort is very little.

Line 395. "longitudinal" -> "meridional"

Line 418. Not only "redistributed". What about the ice growth in the advected parcels? I guess the standard product does not account for that. Does it add a bias in the new product?

Line 423. "Future altimetry missions", as the authors also mention S1 SAR in this bullet point.

Line 429. "As an outlook" -> "In future"

---

## Author Response (AR1)

**Response to Reviewer 1**
We thank the reviewer for his comments and suggestions, which clearly improve the paper. Please find our point-to-point response below.

The manuscript "Drift-aware sea ice thickness maps from satellite remote sensing" by Ricker et al. describes a new algorithm for aggregation of along-track sea ice thickness measurements using corrections from satellite-derived sea ice drift. The produced sea ice thickness dataset is thoroughly validated and supplied with an uncertainty estimate. The manuscript presents new results that are important for climate and cryosphere research. It is well structured and contains all the necessary algorithm details and dataset description details. Nevertheless, a few open questions need to be addressed in the manuscript. I believe it can be recommended for publication after a major revision.

**Major comments**
The text contains grammar mistakes, and a thorough check by a native English speaker is recommended. I marked as much as I could, but I'm not a native English speaker either.
We did another check of the grammar and fixed mistakes accordingly.

It is good that a separate validation of ice drift trajectories used for tracking the parcels is included. However, the presented algorithm (a data-driven model) contains one more module that can potentially significantly affect thickness distribution: accounting for thermal growth. How accurate is that one alone? It seems important to show the RMSEs computed between the input SIT in the parcels (Line 163), and the fitted values (fH, Line 165). It will probably be proportional to the presented growth uncertainty, but RMSE is a more robust method.
The growth correction includes both thermodynamic growth and dynamic changes, as pointed out in the paper. We acknowledge that the applied linear model is coarse, but more complex models may be susceptible to overfitting and reduce the usefulness for estimating sea ice thickness growth. The quality of the estimate basically improves with data density, i.e. spatially coincident parcels from different data acquisition times. The purpose of applying the growth correction is to avoid biases. For a given 25 km grid cell, we typically only find a handful of parcels from different days, and they are usually not symmetrically distributed around the target day (n = 0); for example, we could find SIT estimates from +14 days, +10 days and -2 days. Averaging this set of SIT values likely results in an overestimation of the mean SIT without growth correction. However, the growth correction is rather moderate, as we use a smoothed field (see Fig.4). And the maximum correction period is 15 days. Therefore, we do not expect large changes to the SIT distribution. In the following, we show an example of the difference between corrected and non-corrected SIT for April 15, 2019 (Fig. R1).

The authors used arithmetic mean (Line 207) to compute SIT from advected parcels. Given that the uncertainty of SIT increases with the advection time (both due to growth and drift), would it be more reasonable to use a weighted average with weights inversely proportional to the time of advection? Alternatively, a limit on trajectory length can be added to prevent points from drifting too far (see also comment below on thick ice blocks in MIZ).
In fact, we also considered using a weighted mean. However, we found that this approach does not lead to better results. On the contrary, grid cells that contain only a few parcels are more prone to biases. The weighting increases the risk of primarily relying on one or very few SIT estimates.

It is not clear from the description how the ice drift is used for advecting parcels, i.e., how the ice velocity is integrated to obtain displacement? Is it a linear Euler method or a more accurate Runge-Kutta 4th order, typically used for computing trajectories in inhomogeneous flows? What is the impact on the accuracy of the advection of the selected method (especially in areas with strong ice drift, e.g., Fram Strait).
At this stage, we only use a 1st order Runge-Kutta (RK1) scheme. We acknowledge that a 4th order approach (RK4) is overall preferable. However, given that the drift fields are fairly coarse (75 km resolution) and smooth, daily displacements between roughly 5 and 20 km/day, and a maximum advection period of 15 days, we argue that RK1 is sufficient. This means that the daily displacement is significantly shorter than the size of one grid cell of the drift product. The advantage of the RK1 approach is considerably lower

[Figure]

**Figure R1:** Growth-corrected SIT minus uncorrected SIT for April 15, 2019.

computational costs. But we aim to implement RK4 in the future, especially when using high-resolution SAR-drift products. We have added more details to the description of the drift correction for clarification.

Validation of a non-DA-SIT should be included in section 3.2 Validation with Airborne EM, to be consistent with Section 3.3 Validation with ULS, and illustrate the impact of data awareness. Expectedly, the impact will not be significant, but it needs to be documented.
Yes, we agree. We have added the validation of the non-DA-SIT product and modified Fig. 8 accordingly.

It would be beneficial to homogenize presentation of the comparison – use either 2D histogram (like in Sec. 3.2), or a scatterplot with binning (like in Sec. 3.3). Additionally, the reference data (the independent variable) should be placed on X-axis, and the tested data – on Y-axis, following a common practice.
Agreed. We now use the binning also for Figure 8 and switched the axes.

It is important to add Pearson's correlation or coefficient of determination to the provided RMSE and bias. It's expected to be quite high for the CS2 DA-SIT vs AEM-sit, for example, but nearly zero for Envisat, indicating that it may have low predicting skills for SIT over 2 m.
Agreed. We added Pearson's correlation.

One of the applications of satellite-derived SIT is operational monitoring or assimilation into forecasting models. How efficient can the proposed algorithm be for such an application? Would advection over the last 15 days be sufficient (and important) for providing a more accurate SIT distribution in NRT? Can it be illustrated in the manuscript?
Yes, the product can be operationalized for such applications, i.e. advecting the last 30 or 15 days. If this will improve forecasting models also depends on the model itself and the assimilation. But it will provide a more accurate spatial distribution of SIT on the day of interest compared to traditional monthly means. We have added a sentence mentioning this potential application.

On the video supplement, in mid-January, I noticed some extremely high thicknesses in the MIZ in the Greenland Sea. What is the source of that? It should be either fixed or well presented in the paper with explanations.
Such phenomena might be caused by ocean swell influence on the elevation retrieval. We apply a marginal ice zone filter to avoid such artifacts, but sometimes, affected segments slip through unfortunately.

Figure4. a, left: There are lines with alternating very high and very low growth rates along the CS2 orbits.

Is that an error in the estimation of the growth coefficients or divergence/convergence of sea ice? More light needs to be shed on these cases (e.g., actual time series of advected SIT, divergence estimate from drift fields). It influences a lot the RBF-gridded SIT growth fields at a later stage but doesn't look as convincingly accurate data.

We checked the data particularly for this month. The high growth rate is caused by an individual satellite orbit close to the MIZ during the freeze-up. It might have been affected by ocean swell (see also comment above). This is an issue in the along-track SIT data, which we will try to address in the next version. It also affects the growth estimation, but the effect is small compared to the SIT magnitude of this feature (see also Fig. 5 upper right panel).

**Specific comments**

Line 2. Operational products are averaged over 2 weeks period, "one-month period" need to be rephrased.

It is true that there are products averaging over a two-week period, e.g., the CPOM product offers 14 days gridded SIT data (but also 28 days). However, this leaves significant gaps between the ground-tracks using a 25 km grid. To achieve a complete coverage, interpolation techniques need to be applied. Most of the well-known gridded products are provided as monthly products (e.g., ICESat-2 L4 Monthly Gridded Sea Ice Thickness, AWI CS-2 product, ESA CCI, etc.). Therefore we argue that the typical period is still one month, if a nearly complete coverage shall be achieved. We slightly changed the sentence for clarification.

Line 5. The acronym DA is very often used for Data Assimilation. May authors consider some other abbreviation for the product name to avoid confusing the users who have not read the paper. "Advected SIT" maybe?

We acknowledge the usage of DA for data assimilation. However, we think that within this paper the usage of "DA" should be clear. In presentations and handbooks, DA is typically spelled out, or it is made clear what it stands for.

Line 10. "enables" $->$ "facilitates"
Agreed.

Line 15. "to a state more closely to" $->$ "to a state similar to"
Agreed.

Line 17. "on a large scale"
Fixed.

Line 26. "one month" is not a standard for, e.g., operational products.
Please see our response to a similar comment above. We argue that the majority of the well-known products is provided as monthly grids. Of course, for certain purposes, a two-week coverage can make sense, but to achieve a nearly complete coverage of the entire Arctic, a period of one month is required (e.g., the CS-2 subcycle is one month).

Line 41. "might get lost" $->$ "is neglected"
We rephrased this sentence.

Line 44. "observer" $->$ "observe"
We rephrased the entire paragraph to improve the flow and fixed the typo.

Line 45. Either "single-synchronized orbits" or "a single synchronized orbit"
Agreed. Changed to "single-synchronized orbits".

Line 98. Improve sentence consistency, e.g.: "Location of parcel centres is selected using the NSIDC EASE-Grid 2.0 (Brodzik et al., 2012) with spacing of10 km in both x and y directions."

Thank you for the suggestion, we rephrased the sentence accordingly.

Line 108. What is the uncertainty of the interpolation? Can it be added to the total uncertainty budget in Sec. 2.4?
We neglect the error of the interpolation here as the spatial resolution of the drift product is much lower than the spacing and size of parcels. Therefore, we assume that the drift uncertainty includes any interpolation uncertainty.

Line 116. It is not clear how the position is adjusted. Please clarify in the text.
As explained in our response to the major comments, we used a first-order Runge-Kutta scheme. We have added a paragraph for clarification.

Line 117. "Consequently, subsequent" $->$ "Subsequent"
Agreed

Line 128. $->$ "some parcels have drifted for more than 200 km."
We re-phrased this sentence.

Line 193. "Python"
Fixed.

Line 211. "swapping" $->$ "sliding"
Fixed.

Line 223. How reasonable is the assumption that the uncertainties are uncorrelated? In principle, the same (or very similar) sea ice is sampled by an altimeter in one parcel. With the same roughness that largely contributes to the uncertainty.
We are aware of the fact that in reality the uncertainties are correlated to a certain extent. However, determining the covariances is very difficult as many factors play a role here, not only the roughness, but also the sea surface anomaly interpolation along track, densities, and other variables. The assumption of un-correlated SIT uncertainties is quite common, and also used in the ESA CCI Climate Data Record. For consistency reasons we do the same here.

Line 225. Is the uncertainty of the interpolation procedure taken into account?
No, as mentioned before, we neglect this uncertainty here. The distance between the interpolated and known value is usually smaller than the resolution of the displacements grid.

Line 261. Previously the product was called "drift-aware SIT", which sounds better than "Drift-Awareness SIT" (although, as mentioned earlier a better name can be found to avoid confusion with data assimilation abbreviation). Please check that the product and the algorithm are called consistently.
Thanks for pointing this out. It should indeed be Drift-Aware SIT in that case.

Line 341. The sample sizes for all bins on Fig. 9 should be provided.
Agreed and added to the figure.

Line 359. "Proves" seems to be an overstatement. Fig. 9b indeed illustrates that in some cases, there is a coincident increase of thickness in both products. However, it also shows that in some cases, it is the opposite (e.g., 10 – 20 Dec 2008). For proving, it is crucial to include r, or r2 scores for the comparison of ULS and DA-SIT, as mentioned earlier.
We agree that "Proves" is an overstatement. We rephrased this using the word "indicate".

Line 367. "latitudinal" $->$ "meridional"
Fixed.

Line 373. "on mainly three factors" − > "mainly on three factors"
Fixed.

Line 375. "been"
We fixed this sentence.

Line 383. It is not very clear. Does the conventional product use "parcel positions at the time of the satellite overflight"? Please rewrite.
Yes, exactly. We slightly rephrased this sentence.

Line 386. Quite often "mean difference" is called bias, and "standard deviation of differences" is called RMSD (or RMSE). I would suggest using these terms here and below.
Agreed. We will use the terms bias and RMSD consistently throughout.

Line 390. I would not agree with "In fact, the results for March 2020 appear to be representative overall.", especially for the biases. Maybe this sentence can be simply excluded? As well as the next one, which doesn't bring any useful information.
The statement is related to the RMSD/standard deviation, but we agree that the sentence is misleading. We deleted both sentences as suggested.

Line 392. The peak on Fig. 1.a looks quite sharp, more typical for a Laplace distribution. Is the actual type of the distribution important or just the fact that it is symmetrical? Maybe "Gaussian" can be avoided?
We agree that a Laplace distribution might be a good fit as well. However, the important fact here is that it is symmetric and rather narrow compared to other regions.

Figure 11. Please add bias and RMSE for the entire Arctic.
We have added the bias and RMSD for the entire Arctic.

Line 393. ", the higher the mean difference." − > "", the higher are the mean differences.", or rather ", the higher are the biases."
We changed it to "the higher are the mean differences".

Line 394. Do you mean the Barents Sea instead of the Beaufort Sea? The Barents Sea has the second largest bias, and the mean bias in the Beaufort is very little.
We actually meant Beaufort Gyre, but Barents Sea should be mentioned here as well. We added this accordingly.

Line 395. "longitudinal" − > "meridional"
Fixed.

Line 418. Not only "redistributed". What about the ice growth in the advected parcels? I guess the standard product does not account for that. Does it add a bias in the new product?
No, the standard product does not account for ice growth. However, on a large scale the effect of ice growth is balanced to a large extent if we refer to the mid-month as the reference time in the standard product. With the symmetric ± 15 days scheme, ice growth effects from after and before the target day are opposing. On a regional scale, the effect can be more significant, depending on the distribution and timing of satellite overpasses in the respective areas.

Line 423. "Future altimetry missions", as the authors also mention S1 SAR in this bullet point.
We changed the bullet caption to "Future satellite missions".

Line 429. "As an outlook" − > "In future"
Fixed.

**Response to Reviewer 2**
We thank the reviewer for these helpful comments and suggestions to improve the clarity and flow of the paper. We have addressed and implemented the proposed changes in the paper.

This paper documents the development of a new data production algorithm that considers the drift of sea ice when gridding satellite altimeter measurements of sea ice thickness. This new processing method is a valuable addition to sea ice thickness observations and a timely and important development.
The manuscript is detailed and covers the many intricacies and details involved in its development well, although some of these details can be tricky to follow in the current from. The figures are particularly well presented and well communicate the processing method and impact of incorporating sea ice drift into sea ice thickness gridding. I recommend the article for publication with only minor textual corrections to aid understanding and to make some technical aspects clearer on a first reading – with a single question on the results.
The one result that I am interested to see that was not included in the current draft, is how the linear model used to deal with the variation in thickness over a lag period relates to the mean thickness presented elsewhere. As equation 1 is a function of n, where n=0 is the centre day of a lag window, does this mean p0 will represent the modelled thickness at this centre day whilst considering thermodynamics? How does this p0 relate to the mean of DA measurement stack or the daily data fields? Which of these variables are recommended and what are the best use cases for each? While the final conclusions well cover the context and improvements of the new algorithm, it be will good to also see some recommendations of when best to use the data created here.
I hope the following minor corrections are helpful and aid in the improvement of the manuscript. Harry Heorton

With regard to the linear growth model: For a given 25 km grid cell, we normally only find a handful of parcels from different days, and they are often not symmetrically distributed around the target day (n=0), e.g., we could find SIT estimates from +14 days, +10 days, and -2 days. Averaging this set of SIT values likely results in an overestimation of the mean SIT without growth correction.
Yes, p0 will represent the modeled thickness on the target day (n=0). For those grid cells, where the growth estimation is performed, p0 indeed represents the mean SIT of the growth-corrected SIT estimates. However, we also apply the growth correction to parcels in the vicinity (see interpolation, Fig. 4b), where we cannot derive growth, because we do not have enough spatially and temporally coinciding parcels. We have rephrased the relevant paragraph for clarity.

L 1. This sentence needs to reference 'satellite derived thickness' or similar otherwise the 'along track' part doesn't work.
We rewrote this sentence.

L 9 – 'trackiness' this makes sense for those in the sea ice remote sensing field, but may need expanding for a wider audience
We have rewritten the sentence to avoid "trackiness" in the abstract. We refer to "trackiness" later in the manuscript.

L 16 – summer sea ice extent declined
We clarified this in the text.

L 25. This intro needs the reasons for ice drift to be described – winds and currents.
We added that ice motion is "primarily driven by winds and, to a lesser extent, ocean currents".

L 30 -32. This sentence is out of place for this paragraph and will be better later (L51 and beyond)
We have moved the sentence to the "objectives" paragraph.

L 34 The beginning of the sentence can be removed. Start with "Within the Transpolar drift . . . "

Thanks for the suggestion, we removed the beginning.

L 35 remove 'processes like', they have all been listed!
Fixed.

L 40. Some quantification of these anomalies is needed - I'm not sure which anomaly is being referred to here.
This sentence points to the high along track resolution and small footprint, which enables to even register individual sea ice pressure ridges. We have rewritten this sentence and changed references to clarify.

L 42 – this sentence repeats the previous paragraph and is not needed unless there something specific about the anomalies that is different to thickness measurements.
This sentence shall emphasize the use for sensor merging. We agree that the application is very similar, but we think it is worth to mention the potential use for data merging methods. We rephrased the sentence.

L 44 observer – observe
Fixed.

L45 this whole paragraph needs to be reordered. The details about Cryo2Ice can come first as this will make the rest more clear.
We restructured and rephrased this paragraph.

2.1.1 first paragraph – this general information is not necessarily needed and it may be better just incorporated into the next paragraph. While IceSat2 is mentioned in the intro, the rationale for only using ESA radar altimeters needs to be included. Some info for wider readers on what is meant by level-2 will help here. The crucial aspect for context in wider studies is the choice of SIT over freeboard is needed. This will need a description of the snow data used in the ESA CCI product as this is relevant to the drift of sea ice (for example in the SM-LG product that also considers it – Liston et al (2020).)
We primarily use CS2 and Envisat because this study has been performed within the framework of an ESA-funded project (CCI Sea Ice). However, we also mention at the end of this section, that the method is also applicable to ICESat-2 and other altimeters. We have added a rationale for using ESA altimeters. We also rephrased this section to clarify the meaning of "level-2". We demonstrate the drift-awareness with SIT as this is an end-product that most users are interested in. We clarified this in the text. We also added another reference for further details on the along-track SIT processing of the data used in this study.

L 128 Last sentence needs rewriting – do you mean that some had drifted further than 200km?
We rephrased this sentence.

L 155 this sentence is awkward in the flow of the whole paper. A more coherent sentence or possibly the whole paragraph is that – the collection of drift corrected thickness measurements will represent the same ice during changes to thickness. Thus we use a linear model. The logic at the moments is the other way round – in order to represent the thickness change we must use drift and intersections – this is against the flow from the previous section.
We agree that the explanation lacks clarity. We rephrased and res-structured this paragraph.

L 160. While it is fine to use this resolution this argument neglects important sub-kilometer or floe to floe thickness distributions. Perhaps just state that the method is representative of average ice thickness at this scale length.
We added a sentence for clarification.

Equation 1 While p1 is well discussed, does p0 represent anything physically? Is it related to the mid- or  n=0 day thickness? Does this correspond well to the later arithmetic mean?
p0 represents the thickness at the target day (dt=0).We added a sentence explaining this.

L 169 to calculate the covariance matrix (p0,p1) does the data uncertainty need to be given to the least squares fitting algorithm?
No, it does not at the moment. We will consider other fitting methods that also include the data uncertainties for the next iteration of the algorithm.

L170 while it is very helpful to have this illustrated information within this section, the detailed description of the uncertainty may be more helpful in the results section.
There is no results section as this is more like a method paper. Our aim is to describe the method and the different steps in detail; therefore, we believe the illustration is best placed where it currently appears.

L 204 In any case.. this sentence can be removed and the next be reduced to state the incompleteness windows.
Agreed. We deleted the sentence as suggested.

L 206 Is this missing value issue/method for stacks also true for cells near an advancing/retreating ice edge? How does this all relate to equation 5? Figure 4 shows some peak values near the ice edge in the Chukchi sea, is this related to missing values in the stack and a minimal thickness measurement from CryoSat2? (due to the sensors vertical resolution)
The issue with the incomplete stacks is that we are missing entire satellite passes in certain regions as the altimeter sub-cycle is typically one month. We will also have partly skewed coverage due to a changing ice edge, this is correct. This effect is small compared to the incomplete sub-cycle and also present in conventional monthly gridded products.

Equation 5 – the nanmean approach makes sense for missing values – but does this result in a time bias at the beginning and end of a record – and possibly for advancing retreating ice edge? How does this relate to the earlier comment on equation 1 and p0?
Yes, during the ramping phase in the beginning and at the end, there will be a time bias, because the target day is not in the center of the stack. This is one reason, why we apply the growth correction, which is an attempt to correct for the thickness change of the respective ice parcel between the target day and the time of the actual satellite measurement.

L 218 Is the sigma for each along track taken from the CS2 L2 track files? What variable name is this?
Here we refer to the CCI sea ice thickness product. The L2 files (along-track SIT) contain a variable called "sea_ice_thickness_uncertainty". We added this information in the text.

L 255 is it just the larger uncertainty in drift or just due to the drift being faster? The next sentence is just a repetition – or is there are more detailed reason why the uncertainty is higher near coastlines?
Along the coast lines, higher uncertainties in the passive-microwave drift products are a result of land-contamination and the low resolution. Similarly, in the marginal ice zone, low ice concentration complicates the feature tracking in the ice drift processing.

L 256 - often results.... Is this comment about the paper methods? I'm assuming it is but it will be good to make this certain.
yes, this relates to the methods used in the paper. We tried to clarify this in the sentence.

L 258 – this paragraph all has good information on the drift data. Is there a citation that contains more detailed explanations on the uncertainty? There may not be, these aspects are not always well documented.
Information on uncertainties in the satellite-derived drift product are provided in Lavergne and Down (2023) and Sumata et al. (2014). We added these references in the respective sentence.

L 259 – this sentence needs to be added to the previous paragraph. A figure needs to be cited here. A similar pattern of uncertainty can be seen in figure 2 of Heorton et al (2025).
Agreed, we integrated the sentence into the previous paragraph. Moreover, we added a description on the along-track and total uncertainty, to have a complete description of this figure. We also added the recent

Heorton et al. (2025) study as a references for ice mass balance studies using altimetry.

L 263 this paragraph will benefit from an opening sentence on what metric the buoy data produce to validate the data - a measure of the accumulated uncertainty in parcel location over the lag window. Is this correct? Yes, we use the buoy trajectories as a reference for the DA-SIT trajectories. We re-wrote the sentence.

L 267 'which require' the integration of sea ice thickness measurements over at least.... Agreed and changed.

L 287 about - approximately We assume L 297 is meant here. Agreed.

L 300 – it will be worth repeating here the differences between the AEM total thickness and the thickness of the DA data – I assume this is due to the AEM coming from the snow air interface and representing the thickness of the combined snow and ice thickness? We added a half-sentence clarifying that total thickness is the combination of sea ice thickness and snow depth.

L 329 is this 'conventional gridding' performed by the authors for this study, or data from a prior study? A citation or reference back to the data section is needed here. The 'conventional gridding' was performed by us to make sure that, except of the drift-correction, everything is consistent. We added a reference to Section 2.3.

L 382 – similar to an earlier point, has C-SIT been created for this study? Yes, see above.

Figure 11 – caption needs to say that the SD here is the SD in differences as described in the text. Thanks for pointing this out. We fixed it.

**References**

Heorton, H., Tsamados, M., Landy, J., and Holland, P. R.: Observationally constrained estimates of the annual Arctic sea-ice volume budget 2010–2022, Annals of Glaciology, 66, e9, https://doi.org/10.1017/aog.2025.3, 2025.

Lavergne, T. and Down, E.: A climate data record of year-round global sea-ice drift from the EUMETSAT Ocean and Sea Ice Satellite Application Facility (OSI SAF), Earth System Science Data, 15, 5807–5834, https://doi.org/10.5194/essd-15-5807-2023, 2023.

Sumata, H., Lavergne, T., Girard-Ardhuin, F., Kimura, N., Tschudi, M. A., Kauker, F., Karcher, M., and Gerdes, R.: An intercomparison of Arctic ice drift products to deduce uncertainty estimates, Journal of Geophysical Research: Oceans, 119, 4887–4921, https://doi.org/https://doi.org/10.1002/2013JC009724, 2014.

---

## Author Response (AR2)

**Response to Reviewer**

Line 181. I believe it is important to provide some explanations of the alternating very high and very low growth rates along the CS2 orbits seen in Figure 4 in the manuscript text. It may provide a more comprehensive understanding of the uncertainties associated with the ice growth model.

Yes, we agree that an explanation is helpful in this regard. We added the following paragraph: "In some regions, very high and very low growth rates appear side by side, and overall, the gridded growth estimates exhibit considerable noise. This is partly because the linear fit is highly sensitive to abrupt changes in SIT, which may occur naturally or result from measurement uncertainties. Such variability can introduce significant noise or bias into the estimated growth rates. Additionally, the SIT estimates used for the fit are obtained over differing time intervals, leading to inconsistencies in temporal resolution. These inconsistencies can degrade the stability of the fitted growth estimates."